# Crop Residue Burning and Its Relationship between Health, Agriculture Value Addition, and Regional Finance

Devesh Singh [1,*], Sunil Kumar Dhiman [1], Vijay Kumar [2], Ram Babu [1], Karuna Shree [3], Anjali Priyadarshani [4], Archana Singh [1], Leena Shakya [1], Aparna Nautiyal [5] and Shukla Saluja [6]

[1] Department of Botany, Kirori Mal College, University of Delhi, Delhi 110007, India
[2] Department of Botany, Shivaji College, University of Delhi, New Delhi 110027, India
[3] Department of Geography, Kirori Mal College, University of Delhi, Delhi 110007, India
[4] Department of Zoology, Kirori Mal College, University of Delhi, Delhi 110007, India
[5] Department of Botany, Deshbandhu College, University of Delhi, New Delhi 110019, India
[6] Department of Botany, Sri Venkateswara College, University of Delhi, New Delhi 110021, India
* Correspondence: dev.singh.ece@gmail.com

**Abstract:** Crop residue burning (CRB) poses a serious threat to the climate, soil fertility, human health and wellbeing, and air quality, which increases mortality rates and slumps agricultural productivity. This study conducts a pan-India analysis of CRB burning based on the spatial characteristic of crop residue management practices and analyzes the linkage among health, agriculture value addition, and regional finance using the simultaneous equation to find the causality and panel quantile regression for direct effect and intergroup difference. We discuss some of the alternative crop residue management practices and policy interventions. Along with in situ management, this paper discusses ex situ crop residue management (CRM) solutions. The ex situ effort to manage crop residue failed due to the scarcity of the supply chain ecosystem. Force of habit and time constrain coupled with risk aversion have made farmers reluctant to adopt these solutions. Our results show that financial viability and crop residue have bidirectional causality; therefore, both the central and state governments must provide a financial solution to lure farmers into adopting residue management practices. Our analysis shows that framers are likely to adopt the management solution (farmers have some economic benefits) and are reluctant to adopt the scientific solution because the scientific solution, such as "pusa decomposer", is constrained by the weather, temperature, and humidity, and these parameters vary throughout India.

**Keywords:** stubble-burning; crop residue management; carbon emission; environmental management





## 1. Introduction

India is an agrarian country as 58% of its population depends on agriculture, and it is also the second largest producer of rice and wheat. Due to this reason, it generates a large number of agricultural wastes [1,2]. In most of the states in India, rice, wheat, and maize are the main food crops, mainly in the north Indian states of Gujrat, Maharashtra, Haryana, Rajasthan, Punjab, and Uttar Pradesh [3]. Crop residue burning (CRB) is the result of the mechanization of rice paddy harvesting, which leaves 8 to 10 inches of paddy stalk [4,5]. In some states, a huge quantity of crop residues is left over after harvest, and a large quantity of crop residue is burned across the regions. In the northern states of India, every acre of paddy yields approximately 2.5 tons of stubble [4]. However, in some states, crop residues are often used for biochar, fodder, biofuel production, mushroom cultivation, and energy generation [6].

Residue burning significantly causes an increased amount of particulate matter (e.g., PM2.5) [7]. CRB increases the amount of sulphur dioxide ($SO_2$), volatile organic compounds (VOCs), oxides of nitrogen (NOx), carbon monoxide (CO), and PM, which significantly affect the ambient air quality [8,9]. These particulate matters pose a higher

health risk, monetary losses, and socioeconomic losses [10]. Approximately 62% of the total stubble-burning associated emissions originate from rice and wheat crop residue burning, and 20% are derived from sugarcane. These stubble-burning practices in open fields negatively impact health and cause loss of opportunity cost for farmers (sustainable use of crop residue increases the farmer's income) and loss of agriculture value added to the economy [11].

### 1.1. Why Focus on Crop Residue Management Practices?

India contributes 2.4% of land shares in the world, 18.1% of the global population, and 26.2% of global air pollution, which caused 1.24 million deaths in India in 2017, in which 12.5% of the total deaths were caused by air pollution; in addition, air pollution decreases the life expectancy of people by 1.7 years [12]. CRB increases particulate matter concentration in the atmosphere and adversely impacts the ambient air quality of India, especially in northern states such as Punjab, Haryana, Uttar Pradesh, and Delhi, which possess' a negative health effect on the public due to northwesterly winds, as burnt stubble is transported near human vicinity. Every year north India faces severe air pollution due to CRB in October and November [13]. Health and stubble-burning episodes are directly correlated, and it has been observed that the severity of chronic and non-chronic disease increases during stubble-burning episodes [14]. A survey on health in India showed that crop residue burning increases the fine particles in the air and poses a higher health risk to the public, especially concerning farmers living nearby areas of stubble-burning, which can damage lung function, increase the chances of heart disease, and exacerbate asthma in the short term while in the long term it is associated with reduced lung function and leads to high mortality due to heart disease, increased rate of chronic bronchitis, and lung cancer [10].

The impact of stubble-burning is not limited to human health, soil, and ambient air quality. Stubble-burning has a range of effects on economic growth and causes other social problems such as adverse effects on tourism, agricultural productivity, farmer's socioeconomic condition, and climate effects [15,16]. Another dreadful effect of CRB is that it enhances soil erosion and has a detrimental effect on soil porosity, soil nutrients, soil organic matter, and soil microbes [6]. The overall consciousness is that the total nutrient pool on agricultural land decreases soil fertile quality, which later leads to a decrease in crop production and agriculture value addition to the economy. Further, the heat excreted from the fire of CRB increases the subsoil temperatures at 10 mm depth, nearly from 33.8 to 42.2 °C [17], which affects agricultural productivity directly and indirectly. Further, biomass burning discharges VOCs and NOx in the environment, which forms ground-level ozone (created due to nitrogen oxide and volatile organic compounds in the presence of solar radiation) and affects plant metabolism. This ozone causes serious effects on crops such as wheat and soy, and rice has a moderate effect [18]. Over the last 5 years, India's economic loss estimated due to CRB is nearly USD 1.5 billion [19]. Air pollution causes an increase in operational costs and the loss of 82,000 jobs in the tourism sector [20]. It has been noted that tourism, real estate, construction insurance, and aviation are the most affected industrial sectors. Due to losses bearing on industrial sectors and productivity, the economy per capita of states and nations declined [21]. To discourage CRB practices in India, some states impose a monetary fine on farmers, and continuous practices of CRB in the long term reduce agriculture productivity [22]. It is noted that the total loss of agriculture yield cost due to CRB is reported to be around INR 5000 million per year in India [23]. Therefore, stubble-burning decreases agriculture productivity and its value addition to the country's gross domestic product.

Regional finance, such as "green microfinance", implies providing a formal financial service such as micro-credits, insurance, loans, and digital payment services to the micro-enterprises, communities, and people that previously had no access to these services [24]. Despite this, farmers show reluctance towards these financial services; therefore, the government introduced the direct benefit transfer for not burning crop residues. The main

concept behind the payment for not burning residue is that farmers and communities that are in a position to provide environmental services should be reimbursed for their contribution, and it is another way to provide support to deprived communities. On this concept, more than 300 schemes have been inventoried worldwide [25]. Carbon emission due to crop residue burning is a typical case of poverty and hunger as it comes at the cost of environmental sustainability [26]. Ref. [27] shows that CRB causes losses of livestock which negatively affect the income of farmers. To solve the CRB problem, policymakers suggest that green microfinance motivates farmers not to burn residues, and it is an effective way to reduce CRB practices [28]. Strong green microfinance not just protects the environment but also helps to increase household income. It is evident in developing countries that cash for the environment positively motivates farmers to preserve the environment [29]. The payment for not burning has become one of the most widely accepted tools [30]. According to the government of India, promoting farm machinery banks, financial assistance to registered farmers' societies, co-operative societies of farmers, and self-help groups is the in situ crop residue management solution [31].

Overall, CRB poses a serious threat to the climate, soil fertility, human health and well-being, and air quality which increases mortality rates and slumps agricultural productivity. Figure 1 depict the impacts of crop residue burning and its linkage.

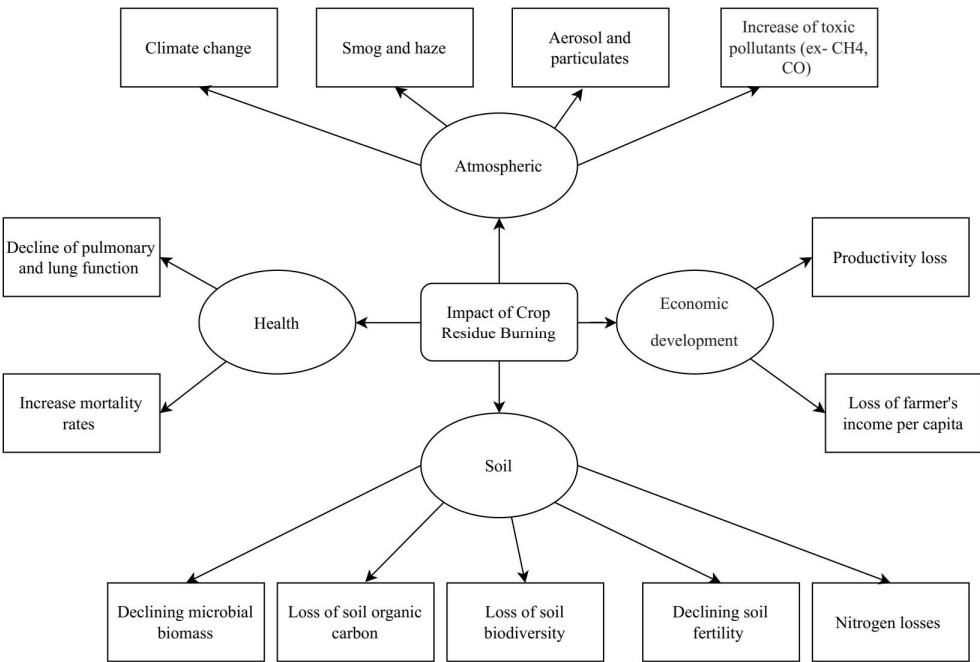

**Figure 1.** Impacts of crop residue burning.

## 1.2. Crop Residue Burning and Ambient Air Quality

The study of ref. [7] presented impact of CRB in Haryana on the Air Quality Index (AQI) of Delhi. Their findings suggest that there is a significant positive association between stubble-burning practices in Haryana and the AQI of Delhi. In solution, their suggestion was to organize large-scale farmer awareness camps for crop residue management (CRM). Similarly, ref. [32] discuss biomass burning and black carbon over Delhi's cloud. Their research results revealed that the biomass burning activities in Punjab and Haryana contribute significantly to Delhi's air pollution. As a result, they suggested an urgency to implement carbonaceous aerosol emission abatement strategies over northern India.

Other researchers [13] discussed residual burning practices using remote sensing. Delhi acquires its pollution from transport, the urban island effect, and stubble-burning practice in Haryana and Punjab, at a height of 500–3000 m added to the long-range transport of smoke particles, which is the main contributor to the worsening of Delhi's AQI. Their conclusions suggest that a detailed study is required to solve this problem from time to time.

Similarly, ref. [33] quantified stubble-burning practices and their impact on north India. The biomass residual is the main contributor to Delhi's increased amount of PM2.5, and it varies between 1% to 58% depending on the atmospheric condition. Their suggestion was that proper air quality improvement strategies are required to improve Delhi's AQI index.

Monitoring of ambient air quality changes in Kaithal, Kurukshetra, and Karnal. These regions have a significant correlation between open stubble-burning and the rise of of SOx, NOx, and PM2.5 concentrations in the atmosphere [33]. However, ref. [32,34] suggested that the biomass burning activities in Punjab and Haryana contribute significantly to Delhi's post-monsoon air pollution. Ref. [35] estimated that the cost of paddy residue burning is INR (Indian National Rupee) 8953 per ha, and the social cost of burning is INR 3199 crores per annum in the region of the northwestern states of India (Punjab, Haryana, Uttarakhand, and western Uttar Pradesh).

To determine the relationship between stubble-burning and $PM_{2.5}$, $PM_{10}$, $NO_2$, CO, and $SO_2$ pollutants near the Delhi regions, researchers employ the Machine Learning (ML) technique [36]. Various meteorological parameters are considered, such as wind speed, temperature, and relative humidity. Their findings show that the ML model performs better in predicting air quality. Similarly, ref. [37] used the ML algorithm for the identification of paddy stubble-burnt fields near the Delhi NCR regions. The results show that ML more efficiently detects the stubble-burning and stubble-burning episodes that occur in the northwestern region of India from the last week of October to the first week of November.

### 1.3. Why Farmers Are Reluctant for Crop Residue's Sustainable Use?

Farmers in the states of Haryana, Punjab, Rajasthan, and Uttar Pradesh grow rice on 10.5 million hectares (26 million acres), generating around 48 million tons of straw in a year, in which about 39 million tons are charred, according to a joint industry–government report [38]. Thus, it has become a major problem in the states of Punjab, Haryana, and western Uttar Pradesh and an extensive problem for sustainable agriculture. This issue began due to the use of mechanized harvesting using combine harvesters which leave behind taller crop residue (about 1–2 ft tall) compared to manual harvesting, where the crops are cut close to the base and leaving stalks less than 6 inches [5,39].

Farmers cannot afford big tractors and machines to plough stubble back into the soil, and the high cost of recycling and settling the crop residue is a major concern for farmers. Therefore, burning is the cheapest, fastest, and easiest method. However, there are other, less harmful ways of clearing agricultural fields, such as the turbo happy seeder (THS) machine, which can uproot the stubble and can then be used as mulch for the field. This solution is not convenient, and farmers are reluctant to hire stubble-removing machines because it is not financially viable for farmers, as most marginal farmers cannot afford them [40].

Farmers are time-constrained between harvesting one crop and sowing the next. Specifically, this could be as small as 7–10 days as this cropping pattern maximizes economic returns. Therefore, farmers do not have sufficient time to treat the crop residue sustainably, especially after wheat and rice harvesting, so they prefer to burn the residue [41]. States, such as Punjab, introduced solutions such as "Happy and Super Seeders". In this solution, the state and central governments facilitate crop residue management machines for farmers in the hope of reducing instances of crop residue burning, but due to insufficient training, logistic management, and delays in the availability of machines, farmers did not adopt the crop residue management machine solution [42]. The India Agriculture Research Institute (IARI) developed a crop residue bio decomposer capsule which decomposes straw within 20 to 25 days. Although the decomposer capsule sustainably decomposes straw, due to time constraints and a lack of information on how to use the decomposer capsule, farmers were scruple for this solution. According to farmers' opinions, a bio decomposer capsule requires certain soil conditions, temperature, and moisture to successfully decompose stubble; therefore, this decomposer capsule failed in certain areas [43].

Along with the in situ solutions, there are also ex situ CRM solutions available. The ex situ effort to manage crop residue failed due to the scarcity of the supply chain ecosystem and dense network of straw banks. Further, according to the Ministry of New and Renewable Energy (MNRE), during the period from 2018–2019 to 2021–2022, more than 39,000 Custom Hiring Centers (CHCs) were identified to deliver 1.95 lakh crop residue management machines in the NCT of Delhi Uttar Pradesh, Punjab, and Haryana [44].

Although the use of straw by end-users increases the agricultural value-added and transporting residue to the end-user, delivery beyond 50 km entails high costs, which is not financially viable for companies engaged in the supply chain. The current decomposition capacity of dismantlers and lack of supply chain and straw demand to the end-user (such as mushroom farmers) is not favoring the current situation [45].

Most farmers have the awareness that CRB destroys the soil's nutrients, but it is not easy to adapt farmers due to their logistics, finances, and complexity; therefore, farmers show reluctance to adopt the current solutions. Generally, farmers are aware of the negative consequences of CRB and intend to adopt sustainable CRM solutions, but due to lack of knowledge, resources, and time constraints, farmers show reluctance to adopt sustainable CRM practices [46]. The overall force of habit and time constrain coupled with risk aversion have made farmers reluctant to adopt CRM solutions [47].

The scientific literature on CRB is classified into three categories based on regional classification: first, single state studies such as Delhi, Punjab, and Haryana [7,48]; second, specific regions such as the northwestern region states of India [34,35,41,49], [50]. Further, there is some other research which focuses on policy-related research papers and presents policy suggestions for northwestern Indian states only; therefore, their contribution to drafting the national policy to crop residue management is limited [9,23,51,52]. Indian agriculture annually produces 501.73 million tons of crop residue, of which 92.81 million tons are burnt. The northwestern states, such as Uttar Pradesh, Punjab, Maharashtra, Madhya Pradesh, and Rajasthan, have a 52.68 million ton residue surplus, and most studies only focus on these regions. Another fact is that around 40.13 million tons of crop residue is unaccounted and to make the Indian agriculture system sustainable, we must focus on a collective approach [53]. It implies that there is an urgency to adopt a collective approach to draft the national policy for residue management. We also observed that there is a limited amount of research available which can be used as pan-India data, and to draft national policy, there is a necessity to analyze the residue burning data nationwide. Therefore, drawing inspiration from the empirical literature gap, this study focuses on nationwide Indian states and attempts to fill the literature gap by achieving the following targets: (i) examine the impact of Indian state's crop residue management capacity on leftover biomass; (ii) study the efficacy of Indian government policies to curb crop residue burning and the related fine particulate matter pollution (PM2.5) in recent years; (iii) suggest a practical solution related to the real world which fits according to Indian conditions to diminish open biomass practices; (iv) find the causality among crop residue burning, increased agriculture value addition, ease of regional finance, and human health; (v) examine the direct effect of agriculture value addition, ease of regional finance, and human health.

## 2. Materials and Methods

### 2.1. Data

This paper used data from the reserve bank of India (RBI) open-source database from 2007 to 2018 and converted it into four panels. This research uses the four variables agriculture value added (AVA) in the economy by states agriculture (in current prices), state-wise credit to agriculture by a scheduled commercial bank (ASCB) in INR, state-wise area of rice and wheat (SAFR) in a thousand-hectare, state-wise number of cases and deaths due to acute respiratory infection (ARI), and the state-wise pattern of agriculture land use (LU) as a gross shown area in thousand hectares. The selection of the time-period and variables was based on the open-source availability of data. Data were collected from the handbook of statistics on Indian states. These data are released by the RBI annually

and Indiastat.com. Each acre of rice and wheat food grains produced 2 tons of crop residue [4,54]. Approximately 22% CRB related carbon emissions come from wheat, and 40% come from rice straw [55]. Therefore, the calculation of residue produced by each state is as follows:

**State-wise Production of crop residue = State-wise Production of Food grains (rice + wheat) in acre X 2.**

Further, we considered crop rotation as a proxy for better crop residue management and spatially aggregated the states, based on high rate of residue management practices. These aggregated states were classified into four groups based on the ratio of area of food grains and land used for agriculture. The states which have high land area but less residue remaining after harvesting have higher index value in this research. Classification of a group of states based on crop residue management was conducted as follows:

**State-wise Area of Food grains (rice and wheat)/agriculture LU, where the index value varies between 1 to 0.**

Classification of states based on CRM:

1. Group-1; Jharkhand, Tripura, Chhattisgarh, Puducherry, Assam, Bengal; Index value 0.9 to 0.5;
2. Group-2; Orrisa, Bihar, Arunachal, Meghalaya, Manipur, Nagaland; Index value 0.5 to 0.4;
3. Group-3; Punjab, Tamil, Andhra, Goa, Uttarakhand, Jammu; Index value 0.3 to 0.2;
4. Group-4; Haryana, Delhi, Karnataka, Himachal, Kerala, Gujrat, Maharashtra, Rajasthan, Mizoram; Index value 0.2 to 0.01.

*2.2. Empirical Model and Specification*

This research used the three-stage least squares (3SLS) simultaneous equation to determine the nexus among the variables, panel quantile regression to examine whether the AVA, ASCB, and ARI are consistently related to the CRB across the differences of 20 quantile levels, and narrative review to confirm the solution and problem that arises from the CRB.

To determine the causality among the variables, we used the simultaneous equation. Analyzing the relationships among the variable is problematic due to the error correlation between variables. Therefore, to overcome this problem, we used the 3SLS simultaneous equation approach because it estimates all parameters in the equation at once and addresses the correlation between error and endogeneity in the number of included equations to be analyzed, which makes the 3SLS method more robust. Since this study uses panel data, the equation in terms of CRB, ASCB, AIR, and AVA is expressed in terms of the panel data equation:

$$(CRB)_{it} = \beta_0 + \beta_1 ASCB_{it} + \beta_2 (AIR)_{it} + \beta_3 (AVA)_{it} + \varepsilon_{it}. \tag{1}$$

$$(CRB)_{it} = \beta_0 + \beta_1 ASCB_{it} + \beta_2 (AIR)_{it} + \beta_3 (AVA)_{it} + \varepsilon_{it}. \tag{2}$$

Taking the log of Equation (1),

$$Ln(CRB)_{it} = \beta_0 + \beta_1 lnASCB_{it} + \beta_2 ln(AIR)_{it} + \beta_3 ln(AVA)_{it} + \varepsilon_{it}. \tag{3}$$

The final econometric of simultaneous equations can be expressed as follows, using Equation (2):

$$Ln(CRB)_{it} = \beta_0 + \beta_1 lnASCB_{it} + \beta_2 ln(AIR)_{it} + \beta_3 ln(AVA)_{it} + \varepsilon_{it}, \tag{4}$$

$$ln(ASCB)_{it} = \beta_0 + \beta_1 ln(CRB)_{it} + \beta_2 ln(AIR)_{it} + \beta_3 ln(AVA)_{it} + \varepsilon_{it}, \tag{5}$$

$$ln(AIR)_{it} = \beta_0 + \beta_1 ln(CRB)_{it} + \beta_2 ln(ASCB)_{it} + \beta_3 ln(AVA)_{it} + \varepsilon_{it}, \tag{6}$$

$$ln(AVA)_{it} = \beta_0 + \beta_1 ln(CRB)_{it} + \beta_2 ln(ASCB)_{it} + \beta_3 ln(AIR)_{it} + \varepsilon_{it}. \tag{7}$$

In the above equations 1, 2, 3, and 4, all variables are explained at the regional level where the subscript i = 1, . . . , N denotes the states, β0 is a constant term for the model equation, β1 − βn is constant terms for associated variables, ε is the error in the equation, and t = 1, . . . , T denotes the time-period.

Further, in this paper, we employed panel quantile regression. This methodology was first introduced by ref. [56]. Quantile regressions provide information about the intergroup difference along with the whole distribution rather than only at the average locations. From a policy angle, it is important to know where the between-group difference is located, and as an example, if financeable viability at the upper end of the distribution is more than at the lower end of the distribution, then policymakers may be able to make informed decisions, rather than pursuing a broad-based approach [57].

Let $Y = X\beta + e$ represent a standard linear model, and we want to study the effect of health, value-added, and the role of financial viability in diminishing the crop residue in India. The linear model calculates the conditional mean based on least-squares, whereas quantile regression evaluates the conditional median. The benefit of using quantile regression is normality; linearity and homoscedasticity are not the concern. It is the extension of linear regression. Therefore, the conditional quantiles function for τth quantile is as follows:

$$Q_\tau (y_{it}) = \beta_0(\tau) + + \beta_1\tau(ASCB)_{it} + \beta_2\tau (AIR)_{it} + \beta_3\tau(AVA)_{it}. \tag{8}$$

In Equation (7), states are indexed by i and time by t, and $y_{it}$ is the crop residue indicator. Where AVA is agriculture value-added in states at the current price, ASCB is state-wise credit to agriculture by a scheduled commercial bank in crores. AIR is the state-wise number of cases and deaths due to acute respiratory infection. All variables were converted into the log and four-panel groups; the details are available in the data section.

### 2.3. Narrative Review

Further, this study used narrative review to determine the implication of CRB and its solution. Further, to find the solution related to the real world, we interpreted the local farmer's opinions published in news articles and reports about the currently available solutions (scientific and CRM). In this research, we followed a protocol to locate all relevant literature using Google Scholar, Scopus, and The Web of Science using the terms stubble-burning, crop residue burning, crop residue management, straw burning, ambient air, agriculture value-added, economic loss due to residue burning, per capita loss, health, mortality, respiratory, lungs, cancer, loss of productivity, nitrogen loss, soil degradation, soil loss, misinformation, sustainable farming, and recycling straw, individually searched and combined with India, Northern India, Delhi, Punjab, and Haryana.

## 3. Results

To determine the causality among crop residue, economic value added to agriculture, health, and financial viability, we used 3SLS, and to check the direct influence and variation along the time, we used panel quantile regression. Before estimating the simultaneous equation model and panel quantile regression models, we investigated the variable's stationarity. We checked the stationarity of the variables through Levin–Lin–Chu (LLC) [58] and Im–Pesaran–Shin (IPS) [59].

### 3.1. Data Description

The descriptive statistics of the used variable after taking the log are shown in Table 1, where correlation, mean, standard deviation, minimum, maximum value, percentile, skewness, and kurtosis are presented. There is no high correlation among the variables, with a maximum 0.6 correlation between AVA and CRB. Further, skewness shows the relative size of two-tail effect, and kurtosis measures the combined two-tail effect. We observed that all variables' kurtosis value were less than 3, which implies that the data are normally distributed.

**Table 1.** Descriptive statistics.

| Variables | (1) | (2) | (3) | (4) | Mean | Std.D. | Min | Max | p1 | p99 | Skew. | Kurt. |
|---|---|---|---|---|---|---|---|---|---|---|---|---|
| (1) CRB | 1.000 | | | | 5.936 | 1.888 | 1.775 | 8.689 | 1.792 | 8.636 | −0.403 | 2.151 |
| (2) ASCB | 0.190 * | 1.000 | | | 8.504 | 2.201 | 4.605 | 12.007 | 4.605 | 11.729 | −0.419 | 1.911 |
| | (0.000) | | | | | | | | | | | |
| (3) AIR | 0.385 * | 0.317 * | 1.000 | | 12.877 | 1.587 | 9.670 | 15.864 | 9.88 | 15.645 | −0.195 | 1.935 |
| | (0.000) | (0.000) | | | | | | | | | | |
| (4) AVA | 0.618 * | 0.324 * | 0.660 * | 1.000 | 13.947 | 1.739 | 10.173 | 16.549 | 10.492 | 16.41 | −0.393 | 1.875 |
| | (0.000) | (0.000) | (0.000) | | | | | | | | | |

\* $p < 0.1$.

We tested the stationarity of the variable LLC and IPS at the levels and the first difference. At the level, all variables were not stationary; therefore, here, we represented the first difference results only. Both LLC and IPS tests are based on the null hypothesis that the variables contain a unit root, against the alternative hypothesis that the panel data series are stationary. Table 2 show the result of the unit root test at the first (I) difference of intercept and trend, where all variables reject the null hypothesis and accept the alternate hypotheses, which implies all variables are stationary at the first difference in both intercept and trend.

**Table 2.** Results of panel unit root test at the first difference I (1).

| | LLC Test | | IPS Test | |
|---|---|---|---|---|
| Variables | Intercept | Trend | Intercept | Trend |
| CBR | −13.526 *** | −12.500 *** | −7.162 *** | −7.245 *** |
| ASCB | −2.52 ** | −6.733 *** | −5.162 *** | −5.162 *** |
| AIR | −11.317 *** | −12.274 *** | −6.701 *** | −7.245 *** |
| AVA | −6.493 *** | −7.893 *** | −6.218 *** | −6.607 *** |

\*\*\* $p < 0.01$, \*\* $p < 0.05$.

### 3.2. Simultaneous Equation Results

Table 3 depict the 3SLS simultaneous results where the Indian regions are categorized into four panel groups based on crop residue management. Group 1 have high crop residue management practices, and Group 4 have the lowest residue management practices. Table 4 show the model accuracy and model performance using $R^2$, RMSE, and $Chi^2$ values. Where RMSE indicates the standard deviation of the residuals (prediction errors), the lower the RMSE better the model. The $R^2$ value represents the efficiency of the models higher the $R^2$ value better the model. The Chi-square value signifies the relationship where the null hypothesis provides no relation between the dependent variables. From Table 4, it is clear that all simultaneous equation models are significant. The causality among the variables is shown in Table 3. In Groups 1, 3, and 4, CRB has a causal effect on AVA, ARI, and ASCB [49,60,61]. Further, ARI have causality with AVA, and ASCB in all four groups. Also, ARI have insignificant relation with CRB in Group 2.From Table 3, we noticed that north Indian states (ex., Punjab, Delhi, Karnataka, Haryana, etc.) are under Groups 3 and 4, achieving this cutoff value during stubble-burning episodes. States specified in Groups 3 and 4 have cultivated lands for rice and paddy farming. In these states, farmers adopted mechanized production farming to increase their profits which led to widespread residue leftovers after harvesting and massive ambient air quality reduction. The result shows that the financial support system can motivate farmers to not burn residues [62].

**Table 3.** Simultaneous equation.

| | Group 1 | | | | Group 2 | | | |
|---|---|---|---|---|---|---|---|---|
| CRB | | 0.884 *** (39.30) | 0.716 * (2.52) | −1.061 *** (−5.91) | | 0.845 *** (23.48) | 1.386 ** (2.60) | 0.257 (1.12) |
| AVA | 1.108 *** (39.30) | | −0.386 (−1.20) | 0.969 *** (4.64) | 1.020 *** (23.48) | | −0.327 (−0.55) | 0.654 ** (2.67) |
| ASCB | 0.103 * (2.52) | −0.0445 (−1.20) | | 0.580 *** (10.86) | 0.0618 ** (2.60) | −0.0121 (−0.55) | | −0.255 *** (−4.86) |
| ARI | −0.334 *** (−5.91) | 0.243 *** (4.64) | 1.263 *** (10.86) | | 0.0489 (1.12) | 0.103 ** (2.67) | −1.087 *** (−4.86) | |
| Cons. | −5.145 *** (−7.16) | 5.139 *** (8.82) | −7.852 ** (−3.00) | 1.991 (1.16) | −8.589 *** (−16.77) | 7.030 *** (16.07) | 17.35 *** (3.65) | 3.791 (1.81) |

| | Group 3 | | | | Group 4 | | | |
|---|---|---|---|---|---|---|---|---|
| CRB | | 0.788 *** (15.51) | 1.775 *** (3.34) | 0.475 *** (3.88) | | 0.749 *** (24.38) | −0.364 * (−2.38) | −0.283 *** (−3.98) |
| AVA | 0.858 *** (15.51) | | −1.354 * (−2.41) | 0.379 ** (2.92) | 1.197 *** (24.38) | | 0.0534 (0.29) | 0.637 *** (8.76) |
| ASCB | 0.0810 *** (3.34) | −0.0567 * (−2.41) | | −0.0683 * (−1.96) | −0.120 * (−2.38) | 0.0110 (0.29) | | 0.379 *** (11.60) |
| ARI | 0.238 *** (3.88) | 0.175 ** (2.92) | −0.751 * (−1.96) | | −0.362 *** (−3.98) | 0.509 *** (8.76) | 1.478 *** (11.60) | |
| Cons. | −9.711 *** (−17.38) | 7.510 *** (12.19) | 26.82 *** (5.42) | 5.152 *** (4.08) | −5.969 *** (−8.00) | 3.410 *** (6.04) | −10.19 *** (−6.91) | 2.622 *** (3.94) |

$*\ p < 0.05$, $**\ p < 0.01$, $***\ p < 0.001$.

**Table 4.** Model summary and performance.

| | Group 1 | | | | Group 2 | | | |
|---|---|---|---|---|---|---|---|---|
| Equation | RMSE | $R^2$ | chi2 | $p$ | RMSE | $R^2$ | chi2 | $p$ |
| CRB | 0.532235 | 0.9271 | 1875.4 | 0.000 | 0.475908 | 0.9032 | 1007.34 | 0.000 |
| AVA | 0.464716 | 0.9294 | 1768.08 | 0.000 | 0.431051 | 0.907 | 1031.52 | 0.000 |
| ASCB | 1.403704 | 0.4536 | 145.46 | 0.000 | 2.293336 | 0.0483 | 29.89 | 0.003 |
| AIR | 0.984931 | 0.382 | 172.76 | 0.000 | 1.106658 | 0.5436 | 119.3 | 0.000 |

| | Group 3 | | | | Group 4 | | | |
|---|---|---|---|---|---|---|---|---|
| CRB | 0.430588 | 0.9212 | 1050.95 | 0.000 | 0.869134 | 0.7895 | 812.55 | 0.000 |
| AVA | 0.407448 | 0.9214 | 1014.66 | 0.000 | 0.698099 | 0.8694 | 1328.03 | 0.000 |
| ASCB | 2.00225 | 0.015 | 13.74 | 0.000 | 1.53559 | 0.5307 | 254.49 | 0.000 |
| AIR | 0.595892 | 0.8153 | 340.58 | 0.000 | 0.790781 | 0.7302 | 520.03 | 0.000 |

Table 4 describe the simultaneous equations model performance, lower the Root Mean Square Error (RMSE), R-square value, Chi-square value, and significant values. The lower the RMSE, the better the model and the higher the R-square value, indicating better model efficiency. AVA had the lowest RMSE value in all four groups, Group 3, and ASCB had the lowest performance.

### 3.3. Direct Effect Estimations Based on Panel Quantile Regression

This paper then set up four panel quantile regression models for Group 1 to Group 4 based on crop residue management practices to examine the direct effects. The coefficient estimations for the 20th, 40th, 60th, and 80th percentiles of the CRB distribution are presented. Panel quantile regression is presented in Table 5, which depicts that the AVA is positively significant in all quantiles and all four groups. The positive coefficient of AVA is an indication of higher agriculture value addition and prosperous economic development. Therefore, the relationship between crop residue and value addition in agriculture is homogeneous. Regarding the ARI coefficients in Group 4, ARI has a negative and significant impact on the 40th, 60th, and 80th percentiles, while it is insignificant and negative in Groups 1 and 2. Health and crop residue are adversely associated. Further, another factor, ASCB, is negatively significant in Group 4 in the 60th and 80th percentiles.

**Table 5.** Panel quantile regression.

| | Group 1 | | | | Group 2 | | | |
| --- | --- | --- | --- | --- | --- | --- | --- | --- |
| | 20th | 40th | 60th | 80th | 20th | 40th | 60th | 80th |
| ASCB | −0.124 | −0.0451 | 0.0619 | 0.0639 | −0.0125 | 0.0173 | 0.0518 | 0.0805 ** |
| | (−1.02) | (−0.42) | (0.79) | (1.31) | (−0.46) | (0.30) | (1.57) | (2.80) |
| ARI | −0.00313 | −0.140 | −0.320 | −0.268 * | −0.000870 | 0.0772 | −0.0230 | −0.0494 |
| | (−0.03) | (−1.00) | (−1.94) | (−2.57) | (−0.02) | (1.30) | (−0.37) | (−0.73) |
| AVA | 1.086 *** | 1.092 *** | 1.084 *** | 1.047 *** | 1.050 *** | 0.988 *** | 1.035 *** | 1.083 *** |
| | (27.82) | (24.28) | (21.50) | (16.89) | (14.63) | (11.05) | (16.33) | (14.74) |
| Cons. | −7.752 *** | −6.471 *** | −4.490 * | −4.404 ** | −8.231 *** | −8.317 *** | −7.681 *** | −7.940 *** |
| | (−7.33) | (−4.17) | (−2.45) | (−2.66) | (−9.17) | (−8.94) | (−9.50) | (−9.90) |
| | Group 3 | | | | Group 4 | | | |
| | 20th | 40th | 60th | 80th | 20th | 40th | 60th | 80th |
| ASCB | 0.0499 | 0.0796 | 0.0604 | −0.0425 | −0.247 | 0.0339 | −0.124 * | −0.203 * |
| | (1.92) | (1.87) | (0.96) | (−1.17) | (−1.79) | (0.76) | (−1.99) | (−2.25) |
| ARI | 0.294 | 0.0873 | 0.188 | 0.188 * | 0.399 | −0.350 *** | −0.213 * | −0.287 * |
| | (1.49) | (0.37) | (1.57) | (2.41) | (1.13) | (−6.94) | (−2.17) | (−2.45) |
| AVA | 0.749 ** | 0.940 *** | 0.860 *** | 0.879 *** | 0.780 *** | 1.087 *** | 1.097 *** | 1.239 *** |
| | (3.19) | (3.89) | (9.33) | (15.38) | (4.77) | (32.05) | (19.73) | (20.66) |
| Cons. | −8.957 *** | −9.057 *** | −8.758 *** | −7.822 *** | −10.07 *** | −5.970 *** | −6.216 *** | −6.084 *** |
| | (−7.57) | (−11.55) | (−12.65) | (−14.52) | (−5.25) | (−9.97) | (−7.19) | (−5.79) |

$* \ p < 0.05, ** \ p < 0.01, *** \ p < 0.001.$

Figure 2 depict the changes in parameter estimates for main variables of financial health and economics, ranging from 0.01 to 0.99 quantiles of the distribution with 95% confidence intervals in the shaded area, and the trended lines represent quantile regression coefficients.

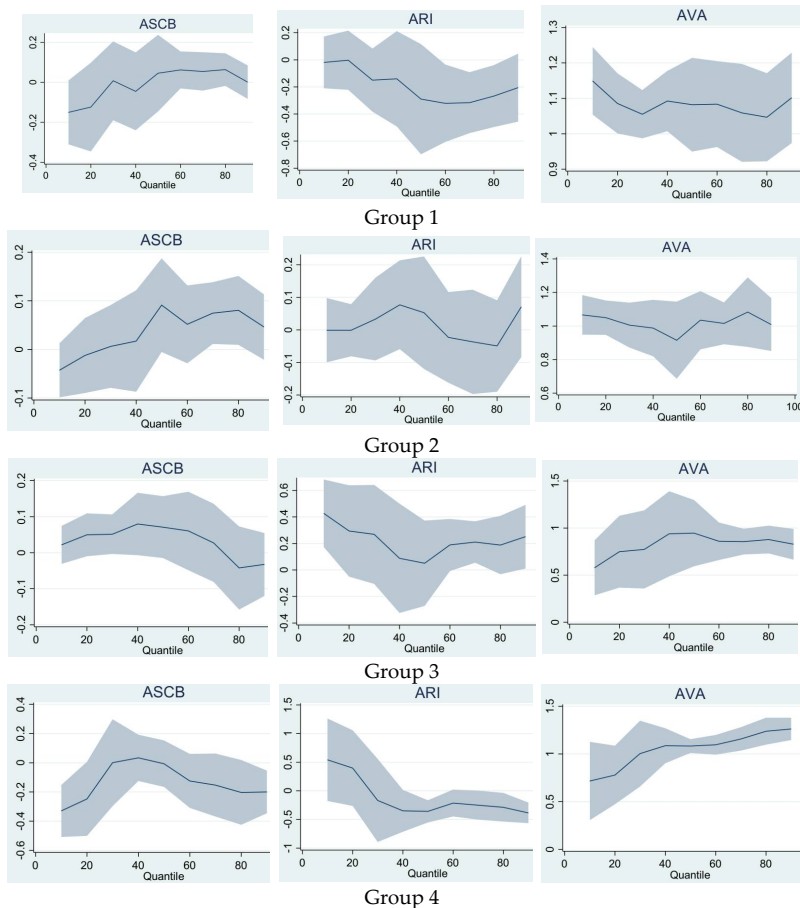

**Figure 2.** Change in panel quantile regressions coefficient.

## 4. Discussion

In India, approximately 0.092 Gigagrams (Gg) of crop residue are burnt every year, with rice straw having the highest contribution of 40% of the total residue burnt, and wheat straw having a 22% share in residue burnt [51]. Our results show that the burning of wheat and rice crop residue significantly contributes to air pollution in Indian regions. Further, these solutions are classified into two groups, scientific and CRM practices. Therefore, this research integrates both management practices and the currently available scientific solutions.

### 4.1. Crop Residue Management Practices

Overall, popular CRM practices among farmers to solve CRB issues are classified into two categories, in situ and ex situ solutions. Some management practices are suggested in the subsequent section.

Biochar: The biomass is converted into biochar using thermal combustion in an oxygen-limited environment. There are some thermochemical methods available which can convert biomass into biochar, such as pyrolysis (decomposition of organic materials in an oxygen-free environment under the temperature range of 250 °C to 900 °C is called pyrolysis), hydrothermal carbonization (the biomass is blended using water and is placed in a closed reactor process can be performed at a low temperature around 180 °C to 250 °C), and torrefaction and flash carbonization (thermochemical conversion process in the absence of oxygen) [63]. Using biochar production, the carbon footprint reduces by 14% during summer rice production and 26% during spring rice production in India [64].

Livestock feed: Traditionally, in India, crop residue is converted into straw and used as animal feed by supplementing it with some additives, but due to high silica content and low digestibility, this practice cannot form a sole ration for livestock. Further, this productive solution is adversely affected by climatic change, mainly due to the absence or non-adoption of effective adaptation and mitigation strategies. Therefore, this approach cannot be prolonged as an efficient, sustainable solution to residue management [27,65].

Biofuel: Production is the process of converting biomass into liquid form using fermentation of crops that are high in starch, which can be blended with existing automotive fuels. Ethanol and biodiesel are two examples of biofuel [66]. Ethanol blending is classified into first-generation, which uses the molasses and byproducts of sugarcane, while second-generation ethanol utilizes biomass and agricultural waste to produce bioethanol. India promotes second-generation bioethanol production to achieve the 20% ethanol blending target [67]. Around 98% of the road transport fuel requirement is currently met by fossil fuels, and the remaining 2% is met by biofuels in India. Currently, 10% of ethanol is blended with petrol. The government of India advanced the target for 20% ethanol blending in petrol (also called E20) to 2025. This will reduce 27 lakh MT of GHG emissions [68,69].

Bacteria and fungi straw decomposition: Decomposition of residue into the soil increases organic matter, water holding capacity, microbial activity, fertility, and soil nutrients. The Pusa bio-decomposer is a recent example used by Indian farmers to deal with CRB issues. This bio-decomposer was developed by the Indian Council of Agricultural Research (ICAR) in 2020. The concentrated form of bio-decomposer is converted into a decomposer mixture (prepare the liquid formulation of bio-decomposer from concentrated solution) using jaggery. It is an efficient and effective technique, as it converts crop residue to valuable bio-compost. However, farmers show reluctance to adopt this decomposer because it takes 8–10 days of preparation and 20–45 days for the degradation process; it is sensitive to the humidity and is not available in all areas [70,71]. Another type of important composting method is the eco-biotechnological process, i.e., vermicomposting involving earthworm, in which crop residue is decomposed and mineralized and converted into vermicompost/vermicast and used as an organic fertilizer [72].

Energy: Regardless of various government policy efforts, approximately 85 to 100 million tons of crop residue are burnt each year. Stubble burns in open fields and wastes a huge amount of energy every year [73]. This stubble can be combusted directly in the presence

of oxygen with other biomass in a combustion chamber to generate heat and convert it into electricity by an electro-mechanical process. Further, the leftover remaining fly ash can be converted into value-added products such as brick and cement manufacturing [74].

Soil nutrient and organic matter: In the long term, frequent residue-burning episodes diminish the microflora and fauna beneficial to the soil and reduce nitrogen, carbon, and a large portion of the other organic matter [51,75]. Furthermore, it is noted that one ton of crop residue-burning diminishes approximately 1.2 kg of sulphur, 5.5 kg of nitrogen, 25 kg of potassium, and 2.3 kg of phosphorous [53]. It is noted that stubble retention balances the potassium and nitrogen content in soil and increases the overall soil organic matter. Stubble retention generates higher grain yields than stubble removal treatments [76]. Figure 3 show the star diagram of the residue management solution suggested by policymakers.

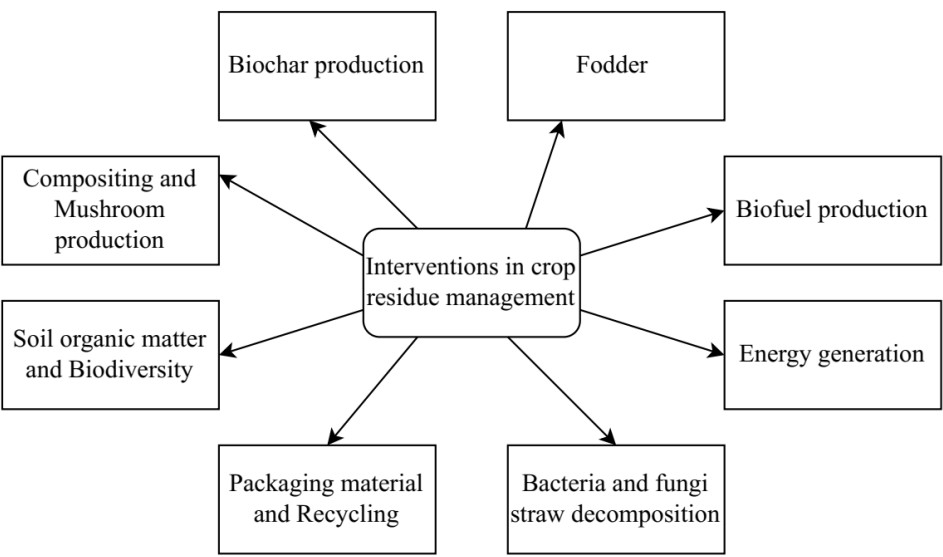

**Figure 3.** Potentially sustainable CRM solution for CRB.

### 4.2. Some Other Scientific Solutions for Crop Residue

Leftover vegetation can be disposed of into value-added products. First, the rice residue was converted into part of the straw using a mechanical process and converted into fine straw. Further, this fine straw was treated with a refiner mechanical pulping (RPM) process with water using hydrothermal treatment (as a media at seven different temperatures from 65 °C to 155 °C) and converted into paperboard, which is used for making various economic value-added products such as food-serving bowls, but this solution is complex and needs ex situ treatment [77].

A study based on engineering intervention provides the scientific solution in the form of a plastic cistern which contains fungal inoculum dose, prepared from *Aspergillus nidulans*, *Aspergillus awamori, Phanerochaete chryosporium,* and *Trichoderma viride*. The researchers found a significant amount of degradation of straw. The treated straw successfully assimilated with the soil and bolstered the nitrogen content [78]. Similarly, ref. [79] found a scientific solution, including bio-processing of rock phosphate using wheat bran-amended paddy straw as a fermentation substrate, which provides a value-added economical soluble fertilizer to enhance phosphorus nutrition in the soil.

The current perspective of stubble-burning studies in Haryana focuses on trying to assess the scale of stubble-burning, discover underlying causes, and the existing condition of different in situ as well ex situ utilization and management practices [80]. The governmental approach to the mechanical management of paddy stubble has also not yielded desired results. The acceptance of mechanical management and the scientific solution was found drastically low due to several factors such as high initial investment, cost of operation, one-time use, unavailability for renting, administrative malpractices, and most

importantly, the time window available. The farmers' opinions suggested that a mix of stubble management techniques should be promoted.

Policy-level interventions such as empowering farmers, promoting the secured market for alternative crops, formalizing land leasing and tenancy agreements, adequate interventions in the existing policy, and introducing a crop insurance system can solve stubble-burning episodes [50]. Another study recommended that the government should promote agri-based food processing units either owned by farmers or public–private partnerships (PPP) to encourage a secured market for alternative crops. These measures would promote crop diversification and discontinue extensive stubble-burning. The government should adopt a more holistic rather selective approach to curb crop-burning across all seasons. Therefore, the government of India must restructure their irrigation policy to diversify the cropping system instead of focusing on only a dual cropping system [81].

### 4.3. Current Policy Interventions to Minimize the Air Pollution

Along with the scientific and management solution, the following policy efforts are initiated by governments to diminish the pollution generated by CRB, which comes under the air prevention and control of pollution act 1981, proposed by the Government of India.

(1) In January 2019, the Ministry of Environment, Forest, and Climate Change (MoEFCC) launched the National Clean Air Programme (NCAP) to prepare clean air action plans to reduce PM2.5 pollution by 20–30% by 2024 (5 years, 700 crores). Outcomes: seven implementation committees, five monitoring committees, and three steerings conducted so far. Recommendation: After increasing the certain level of particulate matter, construction activity should be stopped near the NCR region. Problem: The Preventive measures did not target the root cause.

(2) The Ministry of Environment, Forest, and Climate Change (MoEFCC) launched the PRANA (Portal for Regulation of Air-Pollution in Non-Attainment Cities) information portal. Outcome: A portal for supporting the monitoring of the implementation of NCAP. Recommendation: To spread awareness about stubble-burning and carbon emissions. Problem: The preventive measures did not target the root cause.

(3) A smog tower imported from the US in 2021 with 5000 air filters and 40 fans with estimates of 10 crore budgets. The overall estimated impact of the smog tower was 1km. Outcome: Clean air. Recommendation: Still under investigation. Problem: The preventive measures did not target the root cause, they were high in cost, and were limited to certain areas.

(4) The Delhi government announced a 10-point Winter Action Plan to curb pollution, such as a green war room, with the help of the University of Chicago and a GDI partner (consultant). Outcomes: Monitoring real-time pollution data and mining ISRO image data for stubble-burning. Recommendation: Monitoring and finding the particulate matter hot spots. Problem: The preventive measures did not target the root cause.

(5) The Delhi government assessed the effectiveness of policy interventions for Air Quality Control Regions in the city and suggested $PM_{10}$ and $PM_{2.5}$ can be reduced substantially by increasing the frequency of efficient mechanized cleaning of roads and a sprinkling of water on the roads. Another recommendation was that $CO_2$ emission could be controlled by restricting the entry of commercial vehicles through an alternate path.

## 5. Conclusions

Farmers require a simple solution instead of a complex scientific solution. Therefore, it is better to promote residue management practices which should have the component of economic value to the farmer and also have sound financial viability, sofarmers become more inclusively inclined toward the sustainable use of crop residue management.

Our results show that financial viability and crop residue have bidirectional causality; therefore, both the central and state governments must provide a financial solution such as cash incentives to compensate for the high operating cost of CRM machines, incentivizing the "Happy and Super Seeders" scheme. From the empirical analysis, it is clear that farmers

need an in situ solution which can solve the residue problem within the time frame of 10–15 days. Instead of focusing on the complex scientific solution, policymakers must focus on residue management practices. Our analysis shows that farmers are likely to adopt management solutions (farmers have some economic benefits) but are reluctant to adopt scientific solutions because scientific solutions, such as "*pusa decomposer*", are constrained by the weather, temperature, and humidity, and these atmospheric conditions are varied for each region. For example, the atmospheric conditions in Punjab are completely different from those in Madhya Pradesh. The overall recommendation is that policy efforts to diminish the open residue burning should be tailored according to the region and present available crop residue management capacity instead of "one for all" because the individual farmer land per capita is widely varied in Indian regions from East to West and North to South.

Along with merits, this paper has some research limitations. This study was limited to the used variables and did not consider the "behaviors" or "civic sense" influence on crop residue burning.

**Author Contributions:** Conceptualization, D.S., S.K.D. and V.K.; methodology, D.S.; software, D.S.; validation, D.S., S.K.D., V.K., R.B., K.S., A.P., A.S., L.S., A.N. and S.S.; formal analysis, D.S.; investigation, D.S.; resources, D.S., S.K.D. and V.K.; data curation, D.S.; writing—original draft preparation, D.S.; writing—review and editing, D.S., S.K.D. and V.K.; visualization, D.S.; supervision, S.K.D.; project administration, S.K.D. and V.K.; funding acquisition, D.S., S.K.D., V.K., R.B., K.S., A.P., A.S., L.S., A.N. and S.S. All authors have read and agreed to the published version of the manuscript.

**Funding:** The scholar Devesh Singh is the awardee of ICSSR Post-Doctoral Fellowship. This paper is largely an outcome of the Post-Doctoral Fellowship sponsored by the Indian Council of Social Science Research (ICSSR) Grant File. No. 3-147/2021-2022/PDF/GEN.

**Institutional Review Board Statement:** Not applicable.

**Informed Consent Statement:** Not applicable.

**Data Availability Statement:** Handbook of statistics of India, Reserve bank of India open-source database.

**Conflicts of Interest:** Authors declare no conflict of interest.

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
