# Peer review of "Crop Residue Burning and Its Relationship between Health, Agriculture Value Addition, and Regional Finance"

_atmosphere, doi:10.3390/atmos13091405_

Round 1

Reviewer 1 Report

Thank you for inviting me to review this paper titled “Crop residue burning and its relationship with health, agriculture value addition, and regional finance”. In this manuscript, the authors presented the impact and nexus between crop residue burning, health, agriculture value addition, and regional finance. The authors use a simultaneous equation and panel quantile regression methodology. This manuscript discusses management practices and scientific solutions to solve stubble burning.

·       The first section successfully presented the literature gap and importance of this research.

·       The methodologies, calculations, and assumptions were appropriate.

·       Conclusion section suggests policy implications, recommendations, and limitations.

This manuscript can be considered acceptable if the following minor issues are addressed:

1(1) India achieved a 20% target for ethanol blending in petrol. Therefore, in my suggestion in the discussion section authors have to add the 1st generation and 2nd generation ethanol blending references in the biofuel section.

Overall, the manuscript successfully contributes to the field of crop residue burning and is novel.

Author Response

Respected Editor and Reviewers,

I had been pleased to have an opportunity to revise the manuscript entitled “Crop residue burning and its relationship between health, agriculture value addition and regional finance”. In the revised manuscript, I carefully considered the reviewer’s comments and suggestions. The responses to the concerns raised by reviewers are explained in below “response” section.

All suggestion by reviver is considered and incorporated in the revised manuscript.

Yours Sincerely

Comment: Reviewer 1

India achieved a 20% target for ethanol blending in petrol. Therefore, in my suggestion in the discussion section authors have to add the 1st generation and 2nd generation ethanol blending references in the biofuel section.

Response:

Suggestion to add the research related to ethanol blending in India is added on page number 13.

Thanks very much to the editor and reviewers for your precious suggestions

Reviewer 2 Report

Overall comments and major points:

This paper is well written and emphasizes the importance of managing crop residue combustion by describing its negative impacts, and obtains a two-way causal relationship between the financial viability and crop residue through 3SLS simultaneous equation, and conducts a narrative review approach through much empirical literature to find crop residue solutions and problems of burning,emphasizing the development of economically viable programs to promote crop residue management.The important implication of the paper is to provide a reference for national policy formulation in India and to emphasize the national universality of policy formulation.

Nevertheless, the article has some weaknesses. I highlight two.

1.       Should lines 320-329 of the article be separated from section 2.2? The narrative review seems to be a separate section.

2.       Sections 4.1-4.3 of the discussion section review crop residue management practices, scientific solutions, and current policy interventions, respectively, and may require a more comprehensive review.

Minor points:

Please note some details

1.       The first mention of CRM in line 24 should be followed by the full name.

2.       Figure -1 in line 120 should be Figure 1.

3.       Line 365 ARA may be AVA

4.       Repeat of [39] in line 516.

Hence a revised version should consider the above issues.

Author Response

Respected Editor and Reviewers,

I had been pleased to have an opportunity to revise the manuscript entitled “Crop residue burning and its relationship between health, agriculture value addition and regional finance”. In the revised manuscript, I carefully considered the reviewer’s comments and suggestions. The responses to the concerns raised by reviewers are explained in below “response” section.

All suggestion by reviver is considered and incorporated in the revised manuscript.

Yours Sincerely

Comment: Reviewer 2

  1. Should lines 320-329 of the article be separated from section 2.2? The narrative review seems to be a separate section.

Response: As per the suggestions lines 320-329 are separated and new section 2.3 is created from 339 to 349.

  1. Sections 4.1-4.3 of the discussion section review crop residue management practices, scientific solutions, and current policy interventions, respectively, and may require a more comprehensive review.

Response: Sections 4.1-4.3 are revised in a comprehensive manner as suggested by reviewers. 

Minor points:

Please note some details

  1. The first mention of CRM in line 24 should be followed by the full name.

Response: first mention of CRM is done as per suggestion.

  1. Figure -1 in line 120 should be Figure 1.

Response: done as pointed out by reviewer.

  1. Line 365 ARA may be AVA?

Response: done as pointed out by reviewer.

  1. Repeat of [39] in line 516.

Response: Deleted the reputation.

Thanks very much to the editor and reviewers for your precious suggestions

Reviewer 3 Report

General Comments

The paper presents an important theme. The objectives of this manuscript are consistent with the aims and scope of the Journal. The paper will be of interest to the readers because there are limited studies on this hot issue. The findings of this paper also provide scientific insights to readers and policy-makers to help facilitate plans and policies formulation.

The research can be further improved by taking care of the following suggestions. 

ü  The paper needs revision to correct some linguistic lapses. Language editing can improve the clarity of the work.

ü  I suggest to omit keyword “Aerosol and air quality research”

ü  In the introduction section, please revisit and revise the lengthy sentences. Avoid inappropriate and irrelevant information in this context. I advised the authors to review latest literature.

1-      Understanding farmers’ intentions to adopt sustainable crop residue management practices: A structural equation modeling approach.

2-      Environmental and Health Impacts of Crop Residue Burning: Scope of Sustainable Crop Residue Management Practices

3-      Assessing small livestock herders’ adaptation to climate variability and its impact on livestock losses and poverty

ü  Authors should focus coherence and continuity of the text. Author need to review once again the referred as well as latest articles for proper presentation.

ü  Need to remove unnecessary detail given in the paper. Please be concise.
Need proper citation and justification of your results in empirical results section.

ü  Revisit the conclusion section. Conclusion should be highlighted the main contribution of research to the advancement of science. There are many sentences in the manuscript that are repeated in the same section or in other. Be concise and relevant according to the section.

ü  It would be better to point out the limitations in separate heading after conclusion.

Author Response

Respected Editor and Reviewers,

I had been pleased to have an opportunity to revise the manuscript entitled “Crop residue burning and its relationship between health, agriculture value addition and regional finance”. In the revised manuscript, I carefully considered the reviewer’s comments and suggestions. The responses to the concerns raised by reviewers are explained in below “response” section.

Yours Sincerely

Comment: Reviewer 3

The paper needs revision to correct some linguistic lapses. Language editing can improve the clarity of the work.

Response: Done as suggested

  I suggest to omit keyword “Aerosol and air quality research”

Response: Omitted the keyword “Aerosol and air quality research”

ü  In the introduction section, please revisit and revise the lengthy sentences. Avoid inappropriate and irrelevant information in this context. I advised the authors to review latest literature.

Response: Revised according to the suggestions.

Comment:

1-      Understanding farmers’ intentions to adopt sustainable crop residue management practices: A structural equation modeling approach.

2-      Environmental and Health Impacts of Crop Residue Burning: Scope of Sustainable Crop Residue Management Practices

3-      Assessing small livestock herders’ adaptation to climate variability and its impact on livestock losses and poverty

Response: All three references are cited in the revised manuscript.

ü  Authors should focus coherence and continuity of the text. Author need to review once again the referred as well as latest articles for proper presentation.

Response: Revised according to the suggestions.

ü  Need to remove unnecessary detail given in the paper. Please be concise.

Response: Revised according to the suggestions.

Need proper citation and justification of your results in empirical results section.

Response: Done as suggested by the reviewer

ü  Revisit the conclusion section. Conclusion should be highlighted the main contribution of research to the advancement of science. There are many sentences in the manuscript that are repeated in the same section or in other. Be concise and relevant according to the section.

ü  It would be better to point out the limitations in separate heading after conclusion.

Response: Mentioning the limitations in separate heading will create the redundancy.

Thanks very much to the editor and reviewers for your precious suggestions
